



# 1 Preparation of TFC Membranes Supported with Elelctrospun Nanofibers

# 2 for Desalination by Forward Osmosis

Mustafa Al-Furaiji[1,*], Mohammed Kadhom[2], Khairi Kalash[1], Basma Waisi[3], and Noor Albayati[4]
[1] Environment and Water Directorate, Ministry of Science and Technology, Baghdad, Iraq
[2] Department of Environment, College of Energy and Environmental Sciences, Alkarkh University of Science, Baghdad, Iraq
[3] Department of Chemical Engineering, College of Engineering, University of Baghdad, Baghdad, Iraq
[4] Department of Science, College of Basic Education, University of Wasit, Azizia,Wasit, Iraq
Corrsponding Author: email: alfuraiji79@gmail.com; phone: +964-7736-792-156

## 9 Abstract

Forward osmosis (FO) process has been considered as a viable option for water desalination in
comparison to the traditional processes like reverse osmosis regarding the energy consumption and
economical operation. In this work, polyacrylonitrile (PAN) nanofiber support layer was prepared using
electrospinning process as a modern method. Then, an interfacial polymerization reaction between m-
phenylenediamine (MPD) and trimesoyl chloride (TMC) was carried out to generate a polyamide
selective thin film composite (TFC) membrane on the support layer. The TFC membrane was tested in
FO mode (feed solution facing the active layer) using standard methodology and compared to a
commercially available cellulose triacetate membrane (CTA). The synthesized membrane showed a high
performance in terms of water flux (16 $Lm^{-2}h^{-1}$) but traded the salt rejection (4 $gm^{-2}h^{-1}$) comparing with
the commercially CTA membrane (water flux= 13 $Lm^{-2}h^{-1}$ and salt rejection= 3 $gm^{-2}h^{-1}$) at no applied
pressure and room temperature. Scanning electron microscopy (SEM), contact angle, mechanical
properties, porosity, and performance characterizations were conducted to examine the membrane.

**23 Keywords: Forward Osmosis; TFC membrane; Desalination; Nanofibers; Electrospinning**




## 1. Introduction


Forward osmosis (FO) is an osmotically driven membrane process which uses the difference in osmotic
pressure between the feed solution and a highly concentrated solution (called draw solution) to drive the
pure water from feed solution through the membrane to the draw solution. The FO process has many
advantages over other types of filtration processes such as its low or no trans-pressure, very high rejection
for various contaminants, low membrane fouling tendency than other filtration processes, and easy
building and operating equipment used is very simple and membrane support is less of a problem (Al-
Furaiji et al., 2018; Cath et al., 2006).
One of the crucial aspects of developing FO process is making a suitable membrane for this process. The
ideal membrane has to be highly porous, thin, with good mechanical properties and provides high
rejection of salts and impurities (Ang et al., 2019). Thin-film composite (TFC) membranes have been
widely used in reverse osmosis studies and proven to have excellent performance in desalination (Kadhom
et al., 2016; Kadhom and Deng, 2018). However, recently TFC membranes have attracted more attention
in FO applications.
Commonly, the TFC membranes consist of two layers: a thin selective film that permits water molecules
to pass through but seize salts and other contaminations and a support layer that provides the required
mechanical properties (Ren and McCutcheon, 2014). The selective thin layer is typically prepared by
interfacial polymerization reaction of m-phenylenediamine (MPD) aqueous solution and 1,3,5-
Benzenetricarbonyl trichloride, which is familiarly called trimesoyl chloride (TMC), organic solution on
the support layer. The support sheet is conventionally prepared by phase inversion casting method. Here,
we adopted an emerging technology, electrospinning, to make the support layer. Electrospinning has some
advantages over the traditional phase inversion technique that include producing highly porous layer and
generating sub-micron fibers with highly controllable properties (Waisi et al., 2019). These properties
have led to introduce these nanofiber sheets as promising alternatives for the conventional FO membrane
support layers. Bui and McCutcheon 2013 investigated blending two kinds of polymer (i.e. PAN and
cellulose acetate) to make hydrophilic nanofibers for FO applications (Bui and McCutcheon, 2013).
Huang and McCutcheon 2014 used Nylon 6,6 electrospun nanofibers as support for TFC FO membranes



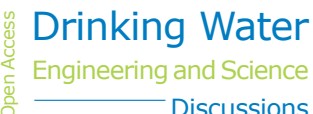
(Huang and McCutcheon, 2014). Chowdhury et al. 2016 prepared and tested a TFC membrane supported
with commercial polyethersulfone (PES) nanofiber membranes (Chowdhury et al., 2017). All these
electrospun nanofiber based TFC membrane showed excellent performance over the commercially
available FO membranes.
In this work, thin film composite polyamide membrane was synthesized by reacting the MPD and TMC
on the electrospun PAN nanofibers support layer and was utilized in the forward osmosis process. The
membranes prepared in this study were mainly characterized by SEM and contact angle to investigate the
impact of the highly porous support layer, in addition to other tests. FO experiments were carried out
using a custom-built setup using sodium chloride as a draw solution for the process.
## 2. Materials and Methods
### 2.1. Materials
Polyacrylonitrile (PAN) of an average molecular weight of 150,000 was purchased from Macklin,
Shanghai, China. N, Ndimethylformamide (DMF) and Isooctane were obtained from Fluka Chemie
AG,Buchs, Switzerlands. The interfacial polymerization raw materials ($m$-phenylenediamine (>99%) and
trimesoyl chloride ( 98%)) were ordered from Merck. Sodium chloride (NaCl) was purchased from
Thomas Baker, India.  Polyethersulfone (PES) of a M.wt. = 150,000 was purchased from Macklin
(Shanghai, China).
The control membrane used in this work was CTA (cellulose triacetate) forward osmosis membrane. This
membrane was provided by Hydration Technology Innovations (HTI) Water Technology (Albany, OR)
and been widely applied for a number of FO applications such as seawater desalination (Linares et al.,
2017), wastewater treatment (Al-Furaiji et al., 2019), and advanced life support systems (Cath et al.,
2005). Properties and images of the membrane can be found elsewhere (McCutcheon et al., 2005).
### 2.2. PAN nanofiber and PES support layers fabrication
PAN nanofibers were prepared using a custom-built electrospinning setup (Figure 1). The electrospinning
setup contained a high voltage power supply, syringe pump, and a rotating drum. Syringe pump was made
from locally available materials. A grounded aluminum rotating drum, which served as a collector, was
placed on a distance of 15 cm from the needle's tip, and an electrical potential was used at a voltage of
30 kV using the power supply device.

Solution of PAN in DMF was prepared by continuously stirring the polymer in the solvent for 24 h at
60ºC. After obtaining the desired solution, it was left to cool and degas overnight at room temperature
prior to electrospinning. The as-prepared polymeric solution was electrospun at flowrate of 1 mL/h onto
an aluminum foil which is peeled off before using the membrane in preparing the TFC membranes.
Electrospinning was conducted at ambient temperature and humidity.

In order to compare the mechanical properties of the prepared support layer with a common

support layer used for the same purpose, a polyethersulfone support sheet was prepared via the phase
inversion phenomenon.  15% PES was dissolved in DMF by applying heat and stirring for 3 h until
colorless solution formed without any polymer residue.  After maintaining the solution at 60 °C during
heating, it was left to cool at room temperature overnight for degassing.  The solution was extended on a
glass plate via a home-made casting knife to a thickness of 130 μm and immersed in a water bath.  The
solution turned to a white sheet and separated form the glass in few seconds.  The sheet was rinsed with
water three times before storing and use.






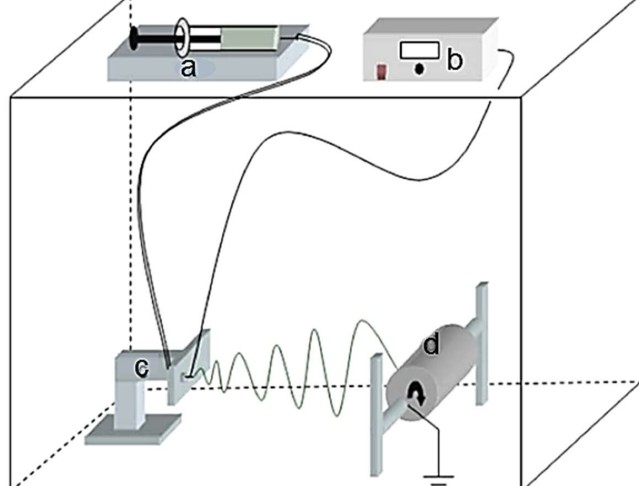


**Figure 1 Photograph of the custom-built Electrospinning setup, (a) syringe pump, (b) high voltage supply, (c) transition stage, and (d) rotating collector.**


### 2.3. Interfacial Polymerization to Make TFC Membrane.

The TFC membranes were prepared via interfacial polymerization reaction at the interface between MPD aqueous solution and TMC organic solution. 2% of MPD was dissolved in DI water to prepare the aqueous solution, while the organic solution was prepared by dissolving 0.15% of TMC in isooctane. The IP reaction was conducted on the PAN support layer as follows: First, the as spun PAN was mounted on a glass plate and the MPD solution poured on its top and kept in contact with the PAN support sheet for 60 s (Kadhom et al., 2016). The excess of the solution was ejected using a squeegee ruler. Then, the TMC solution was poured on the PAN sheet that contains the MPD active sites and kept in contact for 30 s. The resulting TFC membrane was then dried for 10 min at 60 °C and stored in DI water prior to the performance examination.



### 2.4. Membranes Characterizations

The Morphology analysis of the prepared membranes was determined using Scanning Electron Microscope (SEM, VEGA3 - TESCAN, Czechoslovakia). The mechanical properties of the different membranes were obtained from the tensile tests in air at 25 ºC using an Instron microforce tester. A dynamic mechanical analysis (DMA) controlled force module was selected and a minimum of three strips (with a size of 40 mm x 5.5 mm) were tested from each type of membrane. Porosity of the membranes was estimated using gravimetrical method. The membrane was cut as disks with diameter of 2.54 cm (1 in) and weighed ($W_{dry}$). Isopropyl alcohol (IPA) was used as a wetting agent and the membrane weighed after immersed in IPA ($W_{wet}$). The porosity ($\varepsilon$) was calculated from the following equation:

$$\varepsilon = \frac{\left(\dfrac{W_{wet} - W_{dry}}{\rho_{IPA}}\right)}{V} \times 100\%$$

where $\rho_{IPA}$ is the density of IPA and $V$ is the total volume of the sample. Each membrane were tested at least three times. Wettability of the membranes was studied by measuring the contact angle (Theta Lite TL-101 Thailand).

### 2.5. Forward osmosis performance tests

The FO tests were carried out using the experimental set-up illustrated in Figure 2. The installation consists of two tanks: one was specified for the feed solution, while the other was assigned for the draw solution. Both solutions were pumped to the membrane cell using diaphragm pumps from Pure-water®. The membrane was installed in a custom-made cell with dimensions of 3″length, 1″width, and 1/8″depth. The selection of the feed solution and draw solution was according to the standard methodology that was described by Cath et al. 2013. The DI water was used as feed solution while 1 M NaCl solution was used as draw solution. The water permeation flux was estimated as follows:

$$J_w = \frac{\Delta w}{\rho A t}$$

Where $J_w$ is the water flux (Lm$^{-2}$h$^{-1}$), $\Delta w$ represents the difference in the feed solution weight (g), $\rho$ is water density at operating temperature (g/L), A is the actual operative area of the membrane (20 x10$^{-4}$ m$^2$), and t is the experiment's time.



Solute flux through the membrane was estimated by monitoring the conductivity of the feed solution and
using the following equation:
$J_s = \dfrac{\Delta CV}{At}$
Where $J_s$ represents the solute flux (gm$^{-2}$h$^{-1}$), $\Delta C$ is the change in the feed solution concentration (g/L)
(calculated from the conductivity change), and V stands for the volume of feed solution (L).

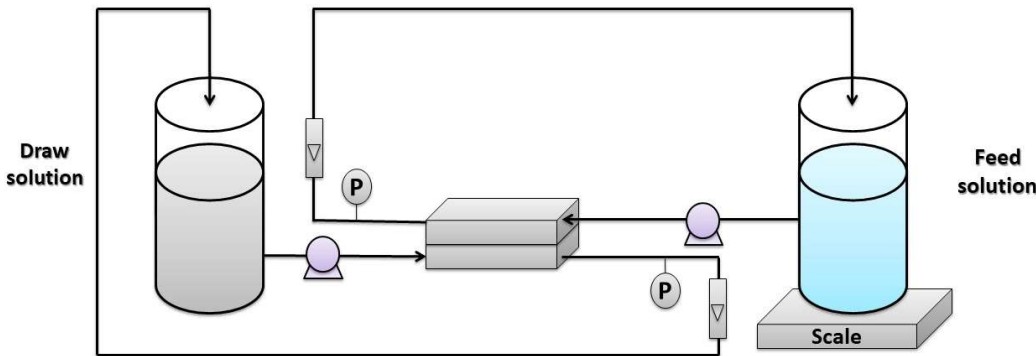


**Figure 2 Schematic diagram of the FO bench-scale test unit.**
## 3. Results and Discussion
### 3.1. Membrane characterization
Figure 3 illustrates the SEM captures of the PAN support layer that was prepared by electrospinning
technique. It can be observed that the membrane structured of smooth and uniform fibers with
approximate diameter of 250 nm. Cross-sectional SEM image (Figure 4) shows that the membrane
consists of nanofibrous layers with a thickness of about 75 microns. It can also be noticed that the
underlying nanofibers owns a very high porosity on their surfaces. This could assure maximum contact
between PAN nanofibers with the draw solution during the forward osmosis operation, which means
higher mass transfer area and consequently higher water flux. Figure 5 illustrates the surface morphology
of the PAN nanofiber membrane after the interfacial polymerization reaction. Also, it can be seen that
polyamide selective membrane was successfully formed on the PAN nanofiber support sheet. Contact

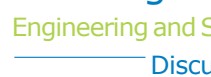

angle measurement of the prepared membranes showed that it has extremely hydrophilic surface with
average contact angle of 35º (Figure 6). Hydrophilicity of membrane's surface is an important factor in
the osmotically driven membrane processes (Darwish et al., 2020). This could be explained as the solutes
can exclusively diffuse within the wetted area of the support sheet. Ultimately, the unsaturated parts inside
the internal structure of the support layer couldn't be calculated as an actual mass transfer area. As much
as the internal surfaces of the pores and inner vacancy get wet, the porous support layer can contribute to
produce a membrane with a better osmotic water flux performance.
**3.2.    Support sheet mechanical properties and porosity**
3.2.1 Mechanical properties
Using a support layer for the TFC membrane that usually applied in nanofiltration, reverse
osmosis, and forward osmosis is inevitable due to the tiny thickness of the active membrane.  The support
layer was found to significantly affect the total performance and commonly made of polymers. Many
factors could influence the layer usage such as its raw material, method and conditions of preparation,
doping additives, porosity, tortuosity, etc (Kadhom and Deng, 2018).  In most cases, the support layer is
manufactured using the phase inversion phenomenon for a low hydrophilicity polymer.  In this work, a
PAN layer was synthesized using the electrospinning, which is expected to produce higher internal
porosity than the sheets produced via phase inversion.  Therefore, the mechanical properties were studied
and compared to the commonly used support layer that produced by phase inversion.
Figure 7 shows the relation between the stress and strain of the PAN sheet.  It can observe that the
maximum stress was 1.258 MPa, which was associated with a strain of 15.31%.  When these values were
compared with 15% Polyethersulfone support sheet (as an example of the familiarly applied support
layers), the stress is lower but the strain is higher.  The measured stress and strain of the PES sheet were
around 2.45 MPa and 8.7%, respectively.  It can be noted that the PAN sheets had a lower mechanical
strength but higher elongation rate.  This result is expected due to the method of preparation, where in the
electrospinning the nanofibers are made individually and connect with each other on the rotating cylinder.
While in the phase inversion, the sheet formed by stiffening the polymer and discarding the solvent.  The


average values of other mechanical properties were listed in Table 1 with the standard deviation of three
measurement values.
**Table 1. Mechanical properties of PAN support layer**

| Mechanical Property | Average value | Standard deviation | Units |
|---|---|---|---|
| Young's modulus | 9.4065 | 1.0288 | MPa |
| Tensile strength | 1.3586 | 0.1428 | MPa |
| Elongation at break | 17.8463 | 3.5857 | (%) |


3.2.2 Porosity
PAN support layer was prepared by the electrospinning to achieve a high porosity. However, the
average porosity values of the PAN and classic PES layers were 92.07% ±2.09 and 60.0% ±1.53,
respectively. From these values, it can be seen that the PAN sheet is more porous than the PES sheet.
This could help in penetrating the water and, anyway, solute through the membrane structure, which could
improve the water flux. Higher porosity means lower unreached spaces and dead ends.

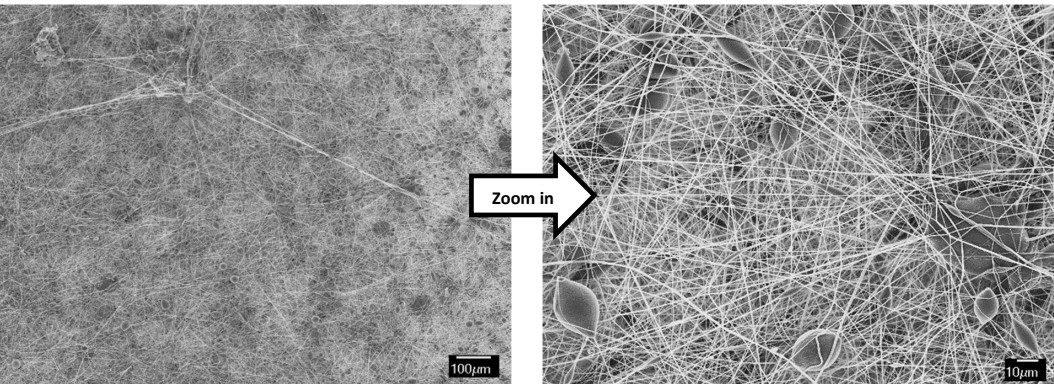


**Figure 3 Surface SEM images of the as-spun PAN nanofiber mat.**

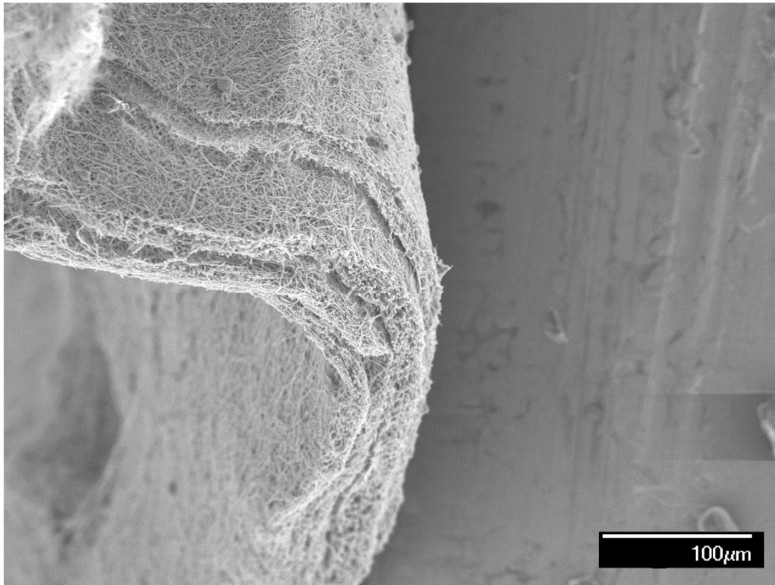


**Figure 4 Cross-sectional SEM image of the as-spun PAN nanofiber mat.**

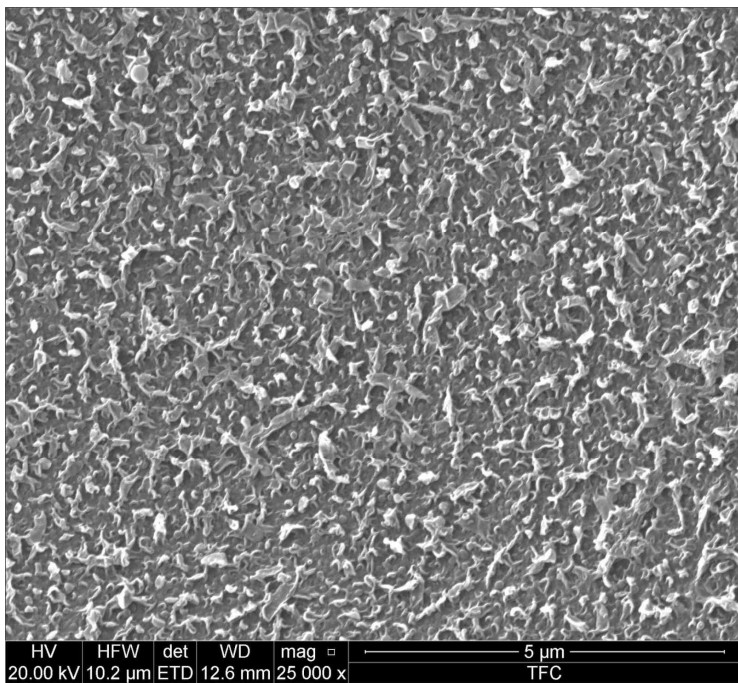

**Figure 5 Surface SEM image of the TFC PAN membrane.**

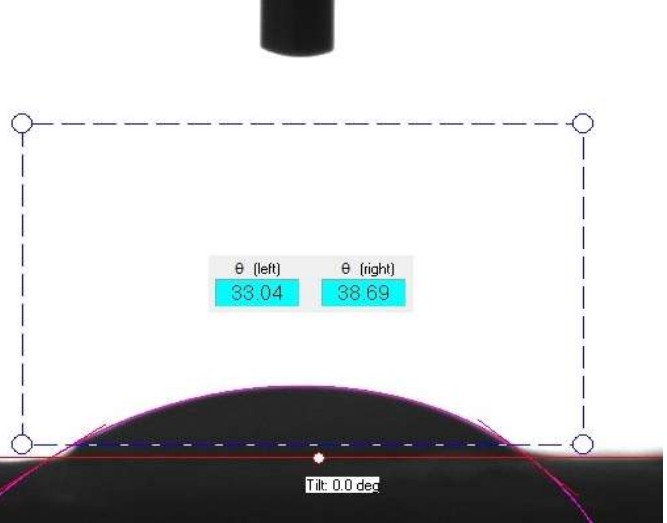

**Figure 6 Contact angle of the PAN nanofiber membrane.**

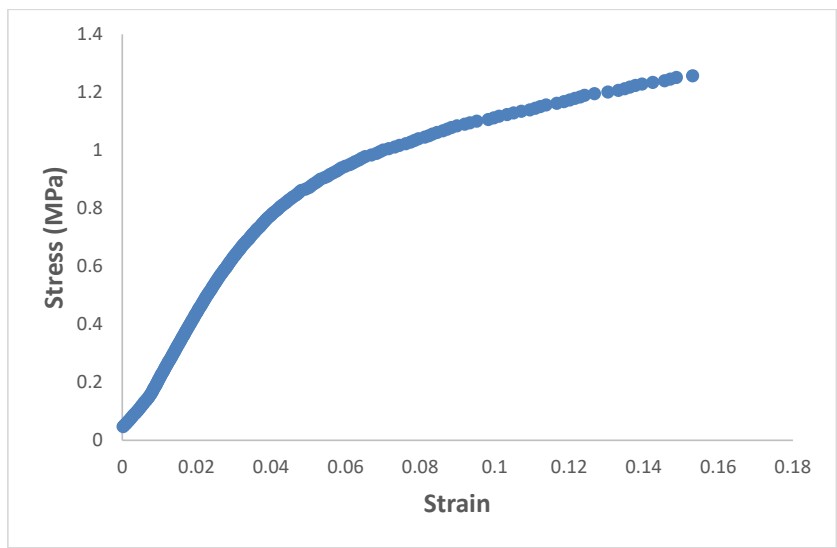


**Figure 7 Stress and strain relationship of PAN support layer**

### 3.3.     Membrane performance in FO operation

The osmotic efficiency of the TFC membrane supported  by nanofiber layer was examined using DI
water as feed solution, whereas 1 M NaCl solution was used as draw solution according to the standard
methodology for testing the osmotically driven membranes (Cath et al., 2013). Results of water flux and
salt reverse flux are clarified in Figures 8 and 9, respectively. PAN-TFC membrane showed stable flux
of about 16 LMH for 20 h of operation. Reverse salt flux exhibited similar behavior with average value
of about 4 GMH. In order to comparing the performance of the PAN-TFC membrane with commercial
membranes, we tested CTA membranes from HTI under the same operating conditions, the results were
illustrated in Figure 10. It can be distinguished from the figure that the PAN-TFC membrane's water
flux was higher than the HTI-CTA membrane's water flux. This could be attributed to the highly porous
surface structure of the nanofiber support layer for PAN-TFC membrane; this porous surface generates
more affective mass transfer area, and consequently higher water flux. However, the reverse salt flux of
the commercial membrane was lower compared to the PAN-TFC membrane. This could ascribe to its
better mechanical strength and rigidity comparing with the nanofibrous composite membranes, which



are commonly have modest mechanical properties. Nevertheless, the FO applications are famous to have
low or no hydraulic pressure required to drive the process; here, it can be resulted that the osmotic
efficiency of the membrane is more important than its rigidity.

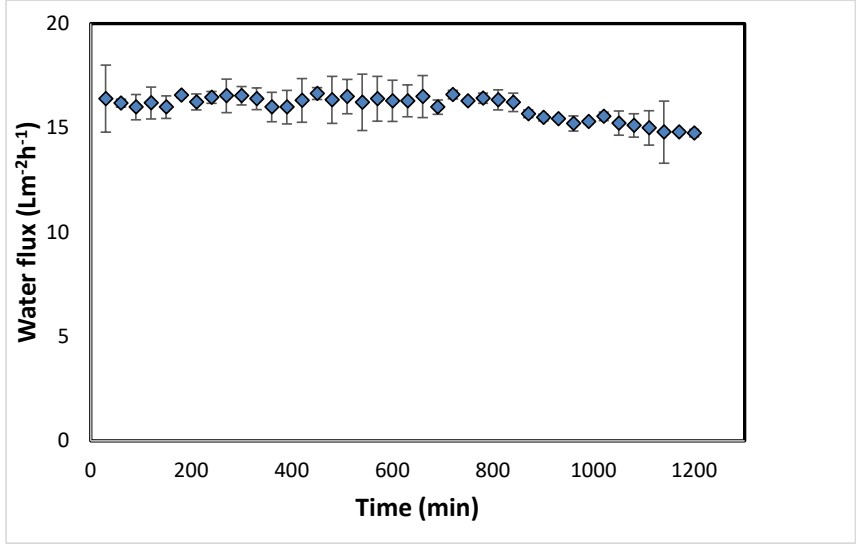


**Figure 8 Forward osmosis water flux for the PAN-TFC membrane. Experimental conditions: feed solution: DI water,**
**draw solution: 1 M NaCl, FO mode, volumetric flow-rate of feed and draw 0.6 L/min, Temp 25º C, zero transmembrane**
**pressure. Results are an average of three experiments with different coupons. Error bars indicate standard deviation.**



**Drinking Water**
Engineering and Science
————— Discussions

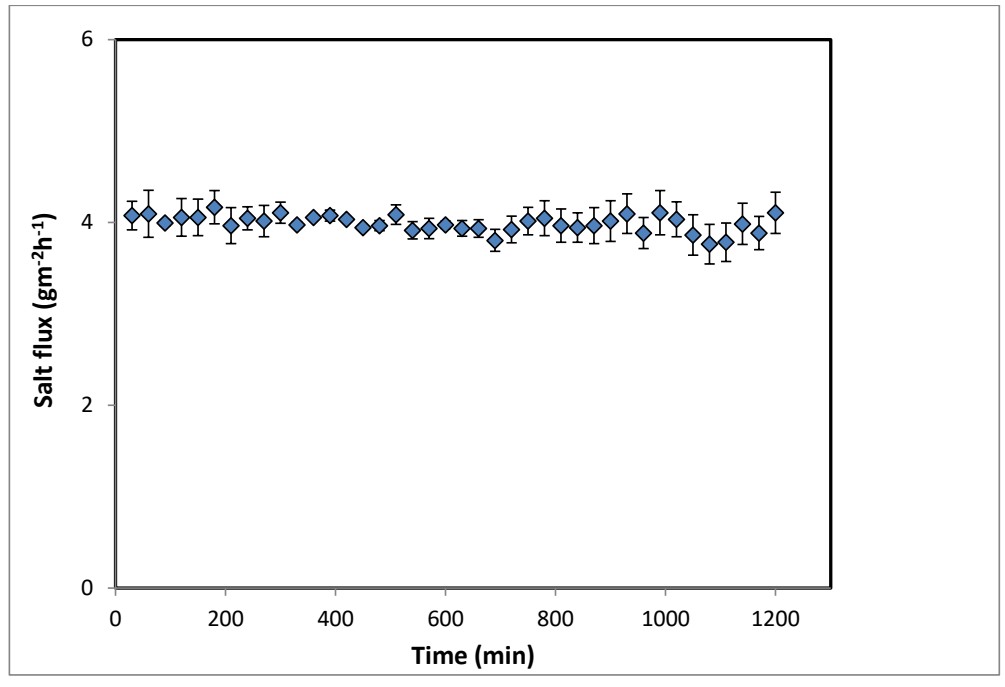


**Figure 9 Forward osmosis salt flux for the PAN-TFC membrane. Experimental conditions: feed solution: DI water, draw solution: 1 M NaCl, FO mode, volumetric flow-rate of feed and draw 0.6 L/min, Temp 25° C, zero transmembrane pressure. Results are an average of three experiments with different coupons. Error bars indicate standard deviation.**

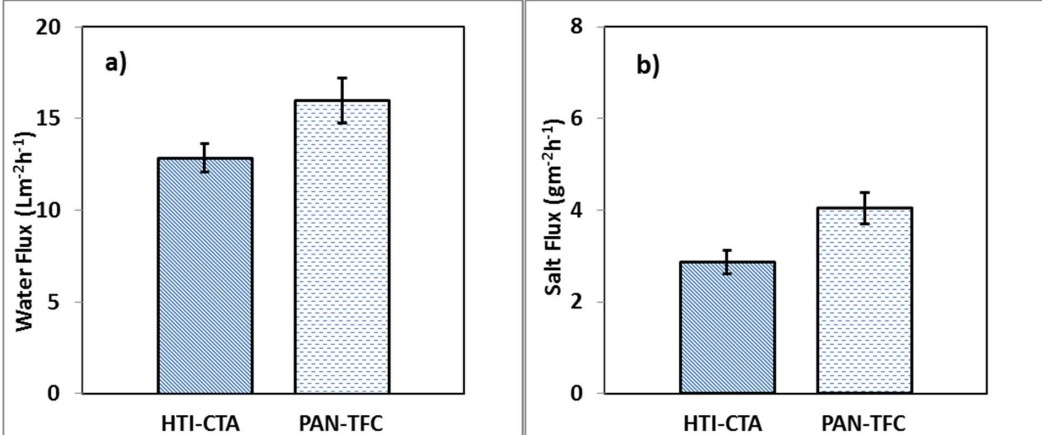


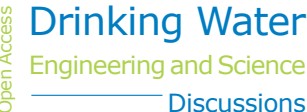
**Figure 10 Forward osmosis water flux and salt flux for the PAN-TFC membrane. Experimental conditions: feed**
**solution: DI water, draw solution: 1 M NaCl, FO mode, volumetric flow-rate of feed and draw 0.6 L/min, Temp 25° C,**
**zero transmembrane pressure. Results are an average of three experiments with different coupons. Error bars indicate**
**standard deviation.**
## 4.  Conclusions and Recommendations
TFC membrane with fibrous structure was prepared in this research and tested for forward osmosis
application. Electrospinning setup was made from locally available parts. This system exhibited stable
operation in making the electrospun nanofiber membrane. The prepared TFC membrane showed good
performance in terms of water flux and salt rejection. TFC-PAN membranes showed a stable water flux
with an average value of 16 LMH comparing to the CTA commercial membranes with 13 LMH water
flux. Future research can focus in incorporating specific nanoparticles to enhance membranes'
performance. Also, studying exposure time of MPD and TMC on the performance of the membrane is
highly recommended.








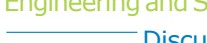

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
