# Peer review of "Preparation of TFC Membranes Supported with Elelctrospun Nanofibers"

_Drinking Water Engineering and Science, 2020_

## Referee Comment (RC1) · Bas Heijman (Referee) · 4 May 2020

line 11 and line 29 suggests that FO is less energy consuming compared to other desalination technologies. This is in fact not true: FO is producing a diluted draw solution. Another treatment technology is required to produce fresh water from the diluted draw solution and to recycle the original draw solution. This downstream technology will be probably RO and will consume about the same amount of energy as conventional desalination. FO as such produces only a useless saline stream.

line 128 mention dimensions in inches. Please use metric units.

line 152: 'Also, it can be seen that polyamide selective membrane was successfully formed...' Please explain how this can be concluded from this photo.

line 154: please justify by references that the contact angle is 'extreemly' low. Also figure 6 can be deleted. Just mention the measured contact angle.
* * *

---

## Short Comment (SC1) · 6 May 2020

Dear reviewer,

Thank you so much for your time and efforts to come up with these valuable comments to improve the quality of our manuscript. The following are our answers to your concerns.

line 11 and line 29 suggests that FO is less energy consuming compared to other desalination technologies. This is in fact not true: FO is producing a diluted draw solution. Another treatment technology is required to produce fresh water from the diluted draw

solution and to recycle the original draw solution. This downstream technology will be probably RO and will consume about the same amount of energy as conventional desalination. FO as such produces only a useless saline stream.

Answer: We agree with the reviewer in that FO process requires another step in recycling the draw solution to become a useful process. However, it is not necessary to use RO process to recycle the draw solution. We used NaCl as a draw solution because it acts as a model draw solution for most of the FO researches and to make a comparison with others works. Different processes –other than RO- can be used to separate the draw solution: 1. Magnetic separation for magnetic nanoparticle draw solutions 2. Heating for NH3-CO2 draw solution 3. Ultrafiltration for nanoparticles-based draw solution 4. Precipitation for Al2(SO4)3 draw solution 5. Nanofiltration for MgSO4 and Na2SO4 draw solutions 6. Membrane distillation for NaCl, KCl, MgCl2 draw solutions Nevertheless, some draw solutions can be used directly after dilution in the FO process such as fertilizer and glucose draw solutions. For more details, please see the following document: • Long, Q.; Jia, Y.; Li, J.; Yang, J.; Liu, F.; Zheng, J.; Yu, B. Recent Advance on Draw Solutes Development in Forward Osmosis. Processes 2018, 6, 165.

line 128 mention dimensions in inches. Please use metric units.

Answer: We changed the dimensions to metric units

line 152: 'Also, it can be seen that polyamide selective membrane was successfully formed...' Please explain how this can be concluded from this photo.

Answer: It can be seen from the SEM image after the IP reaction that it has a leaf-like morphology compared to the PAN support layer which has nanofibrous structure. It has been reported in the literature that the leaf-like structure confirms the formation of the polyamide selective layer.

line 154: please justify by references that the contact angle is 'extreemly' low. Also figure 6 can be deleted. Just mention the measured contact angle.

Answer: The word 'extremely' has been removed from the text. Figure 6 was deleted according to the reviewer's recommendation.

---

## Referee Comment (RC2) · Bas Heijman (Referee) · 14 May 2020

So we agree that forward osmosis is a two step process: first a draw solution is used to draw water trough the FO-membrane and secondly the draw solution has to be concentrated and the fresh water can be harvested. This second step will always consume energy: otherwise it would be possible to build a perpetuum mobile based on harvesting energy from mixing saline water with fresh water. Until now the most efficient and most economical way to recover the draw solution is RO. The author mentions some other possibilities like precipitation (uses a lot of chemicals) and heating for NH3-CO2draw solution. The last option is not applicable because it is very hard to

obtain a very low concentration of NH4+ in the product water and this is mandatory for discharge or even lower concentrations are needed for potable water production. Also there is a problem with dissolving the gasses into the recovered draw solution: A lot of heat is produced when dissolving the gasses. In fact the draw solution should be cooled down before use which will consume a lot of energy. So the biggest problem with FO is that a proper (economical) way to recover the draw solution is not found yet.
* * *

---

## Referee Comment (RC3) · Anonymous Referee #2 · 28 Jul 2020

The manuscript reported the attempt to fabricate TFC FO membrane on an elelctro-spun nanofiber support.

The topic is not new and there have been other studies addressing the use of nanofiber support for TFC FO membranes. Could the authors highlight what is the difference of the reported method as compared to the methods reported in the literature? What would be the advantage of electrospun nanofiber support compared with other nanofiber supports for TFC FO membranes?

The authors compared the lab-scale fabricated FO membranes with the commercial FO membranes. There showed marginal improvement in the water flux and salt rejection

(16 LMH v.s. 13 LMH; 4 GMH v.s. 3 GMH). What would be the potential challenge in scaling up this technology towards a commercial new product? Would scaling-up lead to sacrifice of the performance?

There are numerous FO products in the market. How do your compare the water flux and salt rejection with other commercial FO membranes? Could you cite the figures from literature for comparison?

The strength of the PAN nanofiber support layer has been tested. Have you tested the adherence strength of between the support layer and separation layer? Have you done long-term test on the robustness?

The thickness of support layer is also a crucial factor. A thick support layer will lead to concentration polarization in the support layer, which impairs the performance. Could you compare the thickness of support layer with the commercial products? Would it be feasible to make even thinner support layer with the electrospun nanofiber method?

––––––––––––––––––––––––––––

---

## Author Comment (AC2) · 3 Aug 2020

Dear reviewer, Thank you so much for your time and efforts to come up with these valuable comments to improve our manuscript's quality. The followings are our answers to your concerns. The manuscript reported the attempt to fabricate TFC FO membrane on an electrospun nanofiber support. The topic is not new and there have been other studies addressing the use of nanofiber support for TFC FO membranes. Could the authors highlight what is the difference of the reported method as compared to the methods reported in the literature? Answer: In our paper, we studied the use of a highly hydrophilic polymer (i.e. PAN) in the preparation of a highly porous

nanofiber membrane (support layer) using a home-made electrospinning system that was built from locally available parts. This support layer was tested in FO process, after synthesizing a polyamide thin film composite (TFC) membrane. Its performance was compared to a typical commercial FO membrane (i.e. HTI-CTA membrane). The outcomes of this paper show that highly efficient FO membranes can be prepared in an easy way and also opens the door to investigate different types of other polymers to prepare nanofibrous membranes. Ultimately, we have prepared an inexpensive FO membrane using a very low-price home-made electrospinning system.

What would be the advantage of electrospun nanofiber support compared with other nanofiber supports for TFC FO membranes? Answer: Electrospinning has the ability to produce nanofibers materials with highly tunable properties. Hence, the electrospun nanofibers could be the right candidates for membrane materials for water treatment applications. The support layer of FO membranes should have structural parameters as low as possible (i.e. small thickness, high porosity, and low tortuosity). Electrospun nanofibers can be a good option as a support layer for FO process as they have unique features that matching the properties of the desired support layer. The main advantages of electrospun nanofiber membranes are the easy preparation and highly controllable properties. The authors compared the lab-scale fabricated FO membranes with the commercial FO membranes. There showed marginal improvement in the water flux and salt rejection (16 LMH v.s. 13 LMH; 4 GMH v.s. 3 GMH). What would be the potential challenge in scaling up this technology towards a commercial new product? Would scaling-up lead to sacrifice of the performance? Answer: Although there are many commercial FO membranes in the market, almost all these membranes have not been used in the industrial scale. FO process, in general, still in its early stages in terms of industrial commercialization. Electrospinning process is starting to be used in larger scale to produce commercial electrospun nanofibers membranes for water treatment applications. DuPont manufactured commercial PES electrospun nanofibers and these membranes were tested as support layers for TFC FO membranes (Chowdhury, Huang, and McCutcheon 2017) and for membrane distillation (Al-Furaiji et al.

2019) process. To summarize, electrospinning technique has been already scaled up and commercial products were produced and tested in membrane processes (FO and MD). However, testing these commercial products on larger scale needs further investigations. There are numerous FO products in the market. How do you compare the water flux and salt rejection with other commercial FO membranes? Could you cite the figures from literature for comparison? Answer: We will add the table below to compare the performance of our membranes with the commercially available FO membranes from literature. Membrane Feed Solution Draw Solution Water Flux (LMH) Salt Flux (GMH) Reference PAN-TFC DI 1 M NaCl 16 4 This work HTI-TFC DI 1 M NaCl 15 4.5 (Ren and McCutcheon 2018)

Aquaporin TFC DI 1 M NaCl 9 4 (Xia et al. 2017)

Oasys TFC DI 1 M NaCl 30 50 (Cath et al. 2013)

Porifera CTA DI 1 M NaCl 29 - (Roy et al. 2016)

The strength of the PAN nanofiber support layer has been tested. Have you tested the adherence strength of between the support layer and separation layer? Answer: During the interfacial polymerization reaction between the MPD and the TMC, a very thin polyamide layer is formed on the top of the PAN support layer. Typically, the thickness of the polyamide layer is about 100 nm as reported in our previous paper (Kadhom, Hu, and Deng 2017), while the thickness of the PAN support is about 100 $\mu$m. Measurement of adherence strength between the two layers is not practically possible due to the small thickness of the polyamide layer. However, the performance test proved that the selective layer was kept stick to the PAN support layer at least during the time of the experiment, where the salt rejection maintained high. Having a high salt rejection is impossible without the selective thin film membrane. Have you done long-term test on the robustness? Answer: The prepared membranes in this work was only tested in short-term experiment. However, long-term testing will be considered in our future investigations. Thank you for mentioning this. The thickness of support layer is also a

crucial factor. A thick support layer will lead to concentration polarization in the support layer, which impairs the performance. Could you compare the thickness of support layer with the commercial products? Would it be feasible to make even thinner support layer with the electrospun nanofiber method? Answer: The thickness of our membrane ($\sim$100 $\mu$m) lies within the range of the thickness of the commercially available FO membranes (50-150 $\mu$m). In the electrospinning method, the membrane thickness can be highly controlled and thinner support layer can be easily produced. Here, the manufacturing problems of phase inversion (the common preparation method of RO and FO support layers) are overcome. However, very thin electrospun nanofiber membrane will be difficult to deal with and the robustness of the prepared membranes will not be enough to withstand the testing conditions. So, there is a tradeoff between the concentration polymerization effect and the robustness of the membrane and finding the optimum thickness can be a good topic for future researches.

References Al-Furaiji, Mustafa, Jason T Arena, Jian Ren, Nieck Benes, Arian Nijmeijer, and Jeffrey R. McCutcheon. 2019. "Triple-Layer Nanofiber Membranes for Treating High Salinity Brines Using Direct Contact Membrane Distillation." Membranes 9 (5): 60. https://doi.org/10.3390/membranes9050060. Cath, Tzahi Y., Menachem Elimelech, Jeffrey R. McCutcheon, Robert L. McGinnis, Andrea Achilli, Daniel Anastasio, Adam R. Brady, et al. 2013. "Standard Methodology for Evaluating Membrane Performance in Osmotically Driven Membrane Processes." Desalination 312 (March): 31–38. https://doi.org/10.1016/j.desal.2012.07.005. Chowdhury, Maqsud R, Liwei Huang, and Jeffrey R. McCutcheon. 2017. "Thin Film Composite Membranes for Forward Osmosis Supported by Commercial Nanofiber Nonwovens." Industrial and Engineering Chemistry Research 56 (4): 1057–63. https://doi.org/10.1021/acs.iecr.6b04256. Kadhom, Mohammed, Weiming Hu, and Baolin Deng. 2017. "Thin Film Nanocomposite Membrane Filled with Metal-Organic Frameworks UiO-66 and MIL-125 Nanoparticles for Water Desalination." Membranes 7 (2): 31. https://doi.org/10.3390/membranes7020031. Ren, Jian, and Jeffrey R. McCutcheon. 2018. "A New Commercial Biomimetic

Hollow Fiber Membrane for Forward Osmosis." Desalination 442 (April): 44–50. https://doi.org/10.1016/j.desal.2018.04.015. Roy, Dany, Mohamed Rahni, Pascale Pierre, and Viviane Yargeau. 2016. "Forward Osmosis for the Concentration and Reuse of Process Saline Wastewater." Chemical Engineering Journal 287: 277–84. https://doi.org/10.1016/j.cej.2015.11.012. Xia, Lingling, Mads Friis Andersen, Claus Hélix-Nielsen, and Jeffrey R. McCutcheon. 2017. "Novel Commercial Aquaporin Flat-Sheet Membrane for Forward Osmosis." Industrial and Engineering Chemistry Research 56 (41): 11919–25. https://doi.org/10.1021/acs.iecr.7b02368.

Please also note the supplement to this comment:
https://dwes.copernicus.org/preprints/dwes-2020-9/dwes-2020-9-AC2-supplement.pdf
* * *

---

## Author Comment (AC3) · 11 Aug 2020

Dear Dr. Heijman, We appreciate your comment; we also agree that the FO process still needs further investigation to develop a feasible and economical setup integration. As you mentioned, RO is reliable to produce large amounts of desalinated water, however, it went through many stages to be as it is known today. This included studying the membrane, process, set up, energy consumption, etc. Here, FO needs more investigation to upgrade to the next steps.

Best, Kadhom

[Figure]

Please also note the supplement to this comment:
https://dwes.copernicus.org/preprints/dwes-2020-9/dwes-2020-9-AC3-supplement.pdf

---

## Author Response (AR1)

**Dear reviewer 1,**

Thank you so much for your time and efforts to come up with these valuable comments to improve our manuscript's quality. The followings are our answers to your concerns.

Line 11 and line 29 suggests that FO is less energy consuming compared to other desalination technologies. This is in fact not true: FO is producing a diluted draw solution. Another treatment technology is required to produce fresh water from the diluted draw solution and to recycle the original draw solution. This downstream technology will be probably RO and will consume about the same amount of energy as conventional desalination. FO as such produces only a useless saline stream.

**Answer:** We agree with the reviewer that FO process requires another step in recycling the draw solution to become a useful process. However, it is not necessary to use RO process to recycle the draw solution. We used NaCl as a draw solution because it acts as a model draw solution for most of the FO researches and to make a comparison with others' works. Different processes –other than RO- can be used to separate the draw solution:

1. Magnetic separation for magnetic nanoparticle draw solutions
2. Heating for $NH_3$-$CO_2$ draw solution
3. Ultrafiltration for nanoparticles-based draw solution
4. Precipitation for $Al_2(SO_4)_3$ draw solution
5. Nanofiltration for $MgSO_4$ and $Na_2SO_4$ draw solutions
6. Membrane distillation for NaCl, KCl, $MgCl_2$ draw solutions

Nevertheless, some draw solutions can be used directly after dilution in the FO process such as fertilizer and glucose draw solutions. For more details, please see the following document:

- Long, Q.; Jia, Y.; Li, J.; Yang, J.; Liu, F.; Zheng, J.; Yu, B. Recent Advance on Draw Solutes Development in Forward Osmosis. Processes 2018, 6, 165.

Line 128 mention dimensions in inches. Please use metric units.
**Answer:** We changed the dimensions to metric units

line 152: 'Also, it can be seen that polyamide selective membrane was successfully formed...' Please explain how this can be concluded from this photo.

**Answer:** It can be seen from the SEM image after the IP reaction that it has a leaf-like morphology compared to the PAN support layer which has nanofibrous structure. It has been reported in the literature that the leaf-like structure confirms the formation of the polyamide selective layer.

line 154: please justify by references that the contact angle is 'extreemly' low. Also figure 6 can be deleted. Just mention the measured contact angle.

**Answer:** The word 'extremely' has been removed from the text. Figure 6 was deleted according to the reviewer's recommendation.

**Dear reviewer 2,**

Thank you so much for your time and efforts to come up with these valuable comments to improve our manuscript's quality. The followings are our answers to your concerns.

The manuscript reported the attempt to fabricate TFC FO membrane on an elelctrospun nanofiber support. The topic is not new and there have been other studies addressing the use of nanofiber support for TFC FO membranes. Could the authors highlight what is the difference of the reported method as compared to the methods reported in the literature?

Answer: In our paper, we studied the use of a highly hydrophilic polymer (i.e. PAN) in preparation of a highly porous nanofiber membrane using a home-made electrospinning system that was built from locally available parts. This membrane was tested in FO process and its performance was compared to a typical commercial FO membrane (i.e. HTI-CTA membrane). The outcomes of this paper show that a highly efficient FO membranes can be prepared in an easy way and also open the door to investigate different types of other polymers to prepare nanofibrous membranes.

What would be the advantage of electrospun nanofiber support compared with other nanofiber supports for TFC FO membranes?

Answer: Electrospinning has the ability to produce nanofibers materials with highly tunable properties. The electrospun nanofibers can be the right candidate for membrane materials for water treatment applications. The support layer of FO membranes should have structural parameter as low as possible (i.e. thin, high porosity, and low tortuosity). Electrospun nanofibers can be a good option as a support layer for FO process as they have unique features that match the desired properties of the support layer that was mentioned earlier. The main advantages of electrospun nanofiber membranes are that ease of preparation and the possibility of preparing nanofibers with highly controllable properties.

The authors compared the lab-scale fabricated FO membranes with the commercial FO membranes. There showed marginal improvement in the water flux and salt rejection (16 LMH v.s. 13 LMH; 4 GMH v.s. 3 GMH). What would be the potential challenge in scaling up this technology towards a commercial new product? Would scaling-up lead to sacrifice of the performance?

Answer: Even though there many commercial FO membranes in the market, almost all these membranes have not been used in industrial scale. FO process, in general, still in its early stages in terms of industrial commercialization. Electrospinning process is starting to be used in larger scale to produce commercial electrospun nanofibers membranes for water treatment applications. DuPont manufactured commercial PES electrospun nanofibers and these membranes were tested as support for TFC membranes (Chowdhury, Huang, and McCutcheon 2017) for FO process and for membrane distillation (Al-Furaiji et al. 2019) process as well. To summarize, electrospinning technique has been already scaled up and commercial products were produced and tested in membrane processes (FO and MD). However, testing these commercial products on larger scale needs further investigations.

There are numerous FO products in the market. How do you compare the water flux and salt rejection with other commercial FO membranes? Could you cite the figures from literature for comparison?

Answer: We added a table to compare the performance of our membranes with the commercially available FO membranes from literature.

The strength of the PAN nanofiber support layer has been tested. Have you tested the adherence strength of between the support layer and separation layer?

Answer: During the interfacial polymerization reaction between the MPD and the TMC, a very thin polyamide layer is formed on the top of the PAN support layer. Typically, the thickness of the polyamide layer is about 100 nm as reported in our previous paper (Kadhom, Hu, and Deng 2017) compared to the thickness of the PAN support which is about 100 μm. Measurement of adherence strength between the two layers is not practically possible due to the small thickness of the polyamide layer. However, the performance test proved that the selective layer was kept stick to the PAN support layer at least during the time of the experiment.

Have you done long-term test on the robustness?

Answer: The prepared membranes in this work was only tested in short-term experiment. However, long-term testing will be considered in our future investigations.

The thickness of support layer is also a crucial factor. A thick support layer will lead to concentration polarization in the support layer, which impairs the performance. Could you compare the thickness of support layer with the commercial products? Would it be feasible to make even thinner support layer with the electrospun nanofiber method?

Answer: The thickness of our membrane (~100 μm) fits within the range of the thickness of the commercially available FO membranes (50-150 μm)

In the electrospinning method, the thickness of the membrane can be highly controlled and thinner support layer can be easily produced. However, thinner electrospun nanofiber membrane will be difficult to deal with and the robustness of the prepared membranes will not be enough to withstand the testing conditions. So, there is a tradeoff between the concentration polymerization effect and the robustness of the membrane. Finding the optimum thickness can be a good topic for future researches.

References

Al-Furaiji, Mustafa, Jason T Arena, Jian Ren, Nieck Benes, Arian Nijmeijer, and Jeffrey R. McCutcheon. 2019. "Triple-Layer Nanofiber Membranes for Treating High Salinity Brines Using Direct Contact Membrane Distillation." *Membranes* 9 (5): 60. https://doi.org/10.3390/membranes9050060.

Chowdhury, Maqsud R, Liwei Huang, and Jeffrey R. McCutcheon. 2017. "Thin Film Composite Membranes for Forward Osmosis Supported by Commercial Nanofiber Nonwovens." *Industrial and Engineering Chemistry Research* 56 (4): 1057–63. https://doi.org/10.1021/acs.iecr.6b04256.

Kadhom, Mohammed, Weiming Hu, and Baolin Deng. 2017. "Thin Film Nanocomposite Membrane Filled with Metal-Organic Frameworks UiO-66 and MIL-125 Nanoparticles for Water Desalination." *Membranes* 7 (2): 31. https://doi.org/10.3390/membranes7020031.

Editor's comments

Dear Editor,

Thank you so much for your time and efforts in collecting the reviewers' comments and reading our manuscript. Below are our answers to your comments

Comments to the Author:

Please consider comments of referees to improve your manuscript. Especially the part about newness of the topic is of importance.

We addressed all reviewers' comments in the new version of the manuscript.

Try to minimize the number of Figures. E.g. Figure 6,8,9 do not give much extra information and can be summarized in de text as well.

We appreciate editor's comment; we deleted Figure 6 and included its information in the text. Regarding Figure 8&9, they contain important information about long-term testing of the prepared membrane which is of great importance to confirm the durability of the membrane. So, we suggest, without any offenses, to keep these figures in the paper.

Ask someone (native speaker) to proof read before resubmission

We went through the manuscript again and improved its English.